# Examining Health Disparities and Severity of Depression among Sexual Minorites in a National Population Sample

**DOI:** 10.3390/diseases10040086

**Published:** 2022-10-09

**Authors:** Prashant Sakharkar, Kafi Friday

**Affiliations:** 1College of Science, Health and Pharmacy, Roosevelt University, Schaumburg, IL 60173, USA; 2College of Pharmacy and Health Sciences, Campbell University, Buies Creek, NC 27506, USA

**Keywords:** health disparities, health equity, sexual minority, sexual orientation, gay, lesbian, bisexual, depression, health outcomes, PHQ-9, Patient Health Questionnaire-9

## Abstract

**Background**: Health disparities and mental health issues have not been fully explored among sexual minorities. This study aims to examine health disparities and severity of depression among sexual minorities using a nationally representative sample of the US population. **Methods**: The National Health and Nutrition Examination Survey (NHANES) data from 2011 to 2016 were analyzed. The Patient Health Questionnaire (PHQ-9) was used to examine the severity of depression among sexual minorities compared to heterosexuals. Data were analyzed for descriptive statistics and associations using the Chi-squared test. A multivariate logistic regression analysis was used to quantify the magnitude of association between severity of depression and demographic characteristics. A *p*-value of <0.05 was considered statistically significant. **Results**: Among 7826 participants included, 426 (5.4%) were identified as a sexual minority. Moderately severe to severe depression was observed among 9.3% of sexual minorities with women having higher rates (64.2%) than men. Similarly, sexual minorities were two times more likely to have moderately severe to severe depression, two and half times more likely to see a mental health professional, and one and half times more likely to have genital herpes and be a user of illicit drugs than heterosexuals. In addition, they were less likely to be married and more likely to have been born in the United States, be a U.S. citizen, and earn less than USD 25,000 (*p* < 0.05). **Conclusions**: Sexual minorities are affected by a range of social, structural, and behavioral issues impacting their health. The screening of individuals with depression who are sexual minorities (especially females), illicit drug users, poor, or aged over 39 years may benefit from early intervention efforts.

## 1. Introduction

Differences that affect an individual/group’s attainment of their full health potential and exist among specific population groups are referred to as health disparities. These disparities can be measured by looking at the differences in the incidence, prevalence, mortality, and burden of disease and other adverse health conditions [1]. Although the term health disparity is often used to reflect differences between racial or ethnic groups, disparities do exist across many other dimensions. These dimensions include gender, age, disability status, socioeconomic status, geographic location, and sexual orientation. Health disparities stem from systemic differences (inequities) in the health of groups and communities [2].

According to Healthy People 2020, health equity is the attainment of the highest level of health for all people. Intentional societal attempts to address avoidable inequalities and injustices and eliminate healthcare disparities while valuing everyone equally are required to achieve health equity.

People with non-heterosexual orientations (i.e., lesbian, gay, or bisexual) or people with gender identities such as transgender are termed sexual minorities [3]. Sexual minorities are often referred to by an umbrella term; the acronym LGBT (i.e., lesbian, gay, bisexual, and transgender persons). There is a lack of evidence on the social influences of LGBT people, limited interventional research, and inequities in health care and transgender-specific health needs as indicated in a 2011 Institute of Medicine (IOM) report. This report further outlined the need to advance a research agenda on LGBT health disparities and defined the LGBT population [4].

People (including sexual minorities) who are adversely affected by disparities have systematically experienced greater obstacles to health [3]. Previous studies have reported a higher prevalence of substance abuse, obesity, tobacco use, mental health problems, and lack of access to care among sexual minorities [2]. There are slight differences between sexual minorities and broader society in terms of diseases and health conditions that are more prevalent, while other conditions such as HIV/AIDS disproportionately affect sexual minorities. Health disparities among sexual minorities are influenced by the social determinants of health, and they tend to have an impact over their entire lifespan. Sexual minorities are more likely to be bullied, run away or be forced to leave home, engage in risky sexual behaviors, suffer severe mental health conditions, and even commit suicide [5,6].

Annually, nearly one-fifth of adults report depressive symptoms according to the Centers for Disease Control and Prevention [7]. Sexual minorities report higher rates of depression compared to heterosexuals and are more likely to suffer symptoms of anxiety due to social stigma related to their sexuality [8,9]. Furthermore, the rates of depression vary based on sex, race, and socioeconomic status due to biological and external factors [10,11].

Poor physical and mental health and substance use are more prevalent among sexual minorities. Discrimination and victimization were observed to be among the strongest predictors of poor health outcomes in sexual minority older adults [12,13]. Other predictors such as lack of insurance and financial barriers affect sexual minority women to a greater extant within this group and is a result of the pronounced educational and economic disadvantages they face [14,15]. These disparities reflect the effects of discrimination, stigmatization, and poor general health throughout the life course of sexual minorities [16,17]. Sexual minority older adults were observed to be more likely to have low back or neck pain and weakened immune systems compared to heterosexual older adults [17,18,19]. Meanwhile, sexual minority adolescents to young adults are more likely to be diagnosed with alcohol use disorder than heterosexuals [20]. The risk of conditions affecting the heart and blood vessels, such as cardiovascular disease, was observed to be greater in sexual minorities aged in their mid-40s to 50s relative to their heterosexual counterparts [21]. Even though the risk of cardiovascular disease increases with age, previous findings suggest that this association reflects the heightened LGBT discrimination, chronic stress, and stress-related behaviors that sexual minorities face [22,23]. Additionally, depression was found to be more prevalent among women, racial minorities, and those of lower socioeconomic status in previous studies [11,24,25].

The aim of this study was to examine the health disparities and severity of depression among sexual minorities, compare patient-level factors with heterosexuals, and examine the association of severity of depression amongst sexual minorities using a nationally representative sample of the US population.

## 2. Materials and Methods

### 2.1. Study Population

The data for this study were derived from the 2011–2016 datasets of the National Health and Nutrition Examination Survey (NHANES), a nationally representative sample of non-institutionalized civilian populations in the U.S. [26]. Multiyear data were combined to achieve sufficient sample sizes for statistical analyses. The NHANES is a national survey conducted by the Centers for Disease Control and Prevention that utilizes a combination of at-home interviews and physical examinations to evaluate the health and nutritional status of U.S. residents. The NHANES survey uses a complex, multistage probability sampling design to achieve a nationally representative sample of the U.S. population.

### 2.2. Inclusion/Exclusion Criteria

We elected to include participants aged 18 years and older who completed both the interview and medical examination component of the NHANES survey. The analyses were restricted to participants who completed both the sexual behavior and PHQ-9 questionnaires. Individuals younger than 18 years, those with missing information on the self-reported items in both questionnaires (1958), and those with missing items related to STIs (539) and education (815) were excluded from our analysis.

### 2.3. Sexual Orientation

Sexual orientation was assessed in the NHANES using the following question: Do you think of yourself as heterosexual or straight (attracted to the opposite sex); homosexual or gay/lesbian (attracted to the same sex); bisexual (attracted to men and women); something else; not sure, refused or don’t know? The sexual behavior questions were self-administered in a private room at mobile examination centers (MEC) using an audio computer-assisted self-interview (ACASI) system. The interview was conducted in one of the following languages: English, Spanish, Korean, Vietnamese, or Chinese (traditional/Mandarin, simplified/Mandarin, or traditional/Cantonese). The respondents used earphones to hear questions and also read them on a computer screen. Respondents took their time and used a touch screen to indicate their response. Proxy respondents or translators were not used in situations when the respondents could not self-report. Respondents had the option to select “refuse” or “would rather not answer” if they felt uncomfortable answering, or they could choose not to answer the question at all. Participants self-reporting their sexual orientation as gay, lesbian, or bisexual (GLB) were collectively defined as sexual minorities in this study. A total of 291 (2.9%) participants who responded something else, not sure, refused, or don’t know were not included in the analyses.

### 2.4. Depression

The NHANES uses Patient Health Questionnaire-9 (PHQ-9), a self-reported nine-item questionnaire to assess the severity of depression. The PHQ-9 is a valid and reliable tool for the diagnosis of depressive disorders and severity of depression for clinical and research purposes. The PHQ-9 score ranges from 0 (not at all) to 3 (nearly every day). The presence of none to moderate depression and moderately severe to severe depression was defined by cut-off points from the total score ranging from 0 to 14 and 15 to 27, respectively [27].

### 2.5. Demographic Characteristics

The sociodemographic variables included were age, gender, race/ethnicity, marital status, education, family income, poverty ratio, sexually transmitted infections (STIs), smoking, alcohol use, and illicit drug use. Participants’ age was dichotomized using the median value of 39 years. Race was characterized as Mexican American, other Hispanic, non-Hispanic White, non-Hispanic Black, and other (including Asian, multiracial, and other race). We used the poverty–income ratio (PIR), which is the ratio of family income to the poverty threshold. Using this ratio and the U.S. Census definition of income categories [28], we categorized income levels as a poverty ratio that was poor (<1.35), low income (>1.35–1.84), middle income (1.85–2.99), and high income (≥3.0). In this analysis, marital status was categorized as “married”, “never married”, or “other.” Participants’ smoking status was reported as current smoker; alcohol use as having 4 or more alcoholic drinks every day; and illicit drug use as having ever used heroin, cocaine, or methamphetamine.

### 2.6. Data Analysis

Sociodemographic characteristics were compared using self-reported sexual orientation as the grouping for comparison. We reported the number and percentages for categorical variables and mean and standard error for continuous variables. We used a Chi-squared test for categorical variables to identify statistically significant differences in demographic characteristics. 

We used a logistic regression analysis model to quantify the magnitude of association between severity of depression and sexual orientation. Our selection of covariates for the multivariable regression model was based on the Chi-square analysis. We used age, gender, race/ethnicity, marital status, country of birth, education, income, and insurance as covariates in our multivariate analyses. Taylor series linearization was used for variance estimation. A *p* value of ≤0.05 was considered statistically significant. Analyses were performed using SPSS 28.0 [29] and STATA 14 [30]. SPSS was used for data cleaning, merging, data management, and analyses using unweighted samples. STATA was used for all statistical analyses since it accounts for sampling weights and the complex nature of the sampling design provides statistically valid population inferences. The NHANES analytic guidelines were followed for the creation of multiyear samples.

We did not seek institutional review board (IRB) approval for this study since the NHANES data is de-identified and is publicly available for use.

## 3. Results

We restricted our analysis to respondents aged 18 years and over with valid data on the sexual behavior and depression questionnaire, giving us a final sample of 7826 (78.4%) out of a total of 9975 participants who responded. The sociodemographic characteristics of the sample are presented in Table 1.

A total of 426 (5.4%) participants were identified as a sexual minority, of which 3.4% were women compared to 2.0% men. Participants’ mean age was 39.8 ± 0.37 years; 51.5% were male; the majority were non-Hispanic Whites (65.8%) and had no college degree (66.4%). A significant association was observed between sexual orientation and marital status, country of birth, citizenship, and income level (Table 1). Similarly, a significant association was observed between sexual orientation and genital herpes infection, use of licit drugs (heroin, cocaine, and methamphetamine), and visiting a mental health professional in the past year when conducting Chi-squared analyses (*p* < 0.05) (Table 2).

The estimated proportion of moderately severe to severe depression among sexual minorities was 9.3% with women having higher rates (64.2%) than men. A significant association was observed between all PHQ-9 items and sexual orientation (Appendix A).

Sexual minorities were more than two times more likely to feel down, depressed, hopeless, and bad about themselves, to experience trouble concentrating on things, to report that they moved or spoke slowly or too quickly, and to think they would be better off if they were dead (*p* < 0.05) (Table 3).

Sexual minorities were two times more likely to have moderately severe to severe depression and two and half times more likely to see a mental health professional than heterosexuals (*p* < 0.001). Similarly, sexual minorities were more than one and a half times more likely to have genital herpes and to be a user of illicit drugs than heterosexuals (*p* < 0.05) (Appendix A).

In the multivariate regression analysis, sexual minorities (OR: 1.78, 95% CI: 1.19–2.64; *p* = 0.005) who were older than 39 years (OR: 1.47, 95% CI: 1.18, 1.82, *p* = 0.001), female (OR: 2.06, 95% CI: 1.64–2.58; *p* = 0.001), and not married (OR: 1.23, 95% CI: 1.13, 1.35, *p* < 0.001) and who had less than a high school education (OR: 0.76, 95% CI: 0.67, 0.87, *p* < 0.001), earned less than USD 25,000 (OR: 0.60, 95% CI: 0.53, 0.63, *p* < 0.001), or were born in the US (OR: 0.50, 95CI: 0.38, 0.67, *p* < 0.001) were independently associated with moderately severe to severe depression (Table 4).

## 4. Discussion

Sexual minorities reported moderately severe to severe depression and greater odds of having an STI and being an illicit drug user. The national prevalence of poor health behavior and sexual and social outcomes among sexual minorities was much higher than among heterosexuals. Our results found that genital herpes infection was reported at higher rates among sexual minorities compared to heterosexuals. One study that examined demographic and clinical characteristics of patients found that STIs were more common among sexual minorities. In particular, gay men in this study had a higher prevalence of extragenital gonorrhea and chlamydia infection compared to heterosexuals, in addition to the higher prevalence of genital herpes observed in our study [31].

Sexual minorities were more likely to experience severe depression. Previous studies examined the social and psychological determinants of health and psychological well-being among Americans between the ages of 25 and 74 years and concluded that there was a greater elevation in rates of mental health morbidity among sexual minorities and that they were more likely to meet the diagnosis criteria of severe depression [32].

About 19% of sexual minorities and 44% of lesbians in our study were more likely to see a mental health professional. A study by Hirsch and colleagues reported a much lower estimate of 6% of lesbian women visiting a psychiatrist [33]. These findings further consolidate the fact that mental health and severe depression are significant health issues among sexual minorities, especially lesbians, to varying degrees.

Studies assessing the prevalence of substance use behaviors and substance dependence in sexual minorities found that those who identified as gay, lesbian, or bisexual or reported same-sex attraction had higher levels of substance use and substance dependence (illicit drugs, cigarettes, and alcohol) than heterosexuals [34,35]. Lesbians had greater odds of reporting marijuana use, other drug use, alcohol dependence, and other drug dependence compared to heterosexuals. However, the odds of substance use and substance dependence in gay men did not differ from those of heterosexuals. Bisexual men had more than four times greater odds than heterosexuals of reporting alcohol dependence and other drug dependence. Sexual minorities were found to be more likely to be illicit drug users in our study.

Demographic characteristics among sexual minorities in our study differed significantly compared to heterosexuals. Being female, born in the US, a US citizen, single, poor, and older than 39 years and having less than a college education make one more likely to identify as a sexual minority, which is consistent with the findings reported by Martin-Storey and colleagues [36]. Our findings suggest that females are more likely to identify as lesbian (sexual minorities) than males are to identify as gay. Moreover, they were found to be two times more moderately severe to severely depressed than their male counterpart. Our results are also consistent with the findings of a meta-analysis by Wittgens and colleagues, where lesbian/gay individuals showed a two-times higher risk of depression and anxiety disorder [37]. However, these finding contrast with those of a recent study where the odds of depression among lesbians were reported to be one and half times greater than among gay men [38].

Our results are also consistent with the findings of other studies that observed that sexual minorities were less likely to have earned a college degree. However, in a study by Schuler and colleagues that was conducted using results from the 2015–2018 National Survey on Drug Use and Health, it was found that bisexual females were the most educationally disadvantaged compared to their heterosexual counterparts [39]. Meanwhile, gay males were found to be more likely to have a college degree than heterosexual males [39]. This finding can be attributed to the higher rates of poverty experienced in bisexual women compared to heterosexuals reported in the 2014–2017 Behavioral Risk Surveillance System.

Sexual minorities in a cross-sectional study among 264 gay, lesbian, bisexual, and heterosexual individuals conducted in Serbia were found to suffer from severe depression. In this study, sexual minorities reported significantly more symptoms of depression and suicide attempts compared to heterosexuals. They further reported severe depression as well as higher PHQ-9 scores in those who identified as a sexual minority [40]. Homosexual participants in this study had 27 times higher odds and bisexual participants had six times higher odds of suicidal attempts than heterosexual respondents [40]. These higher odds of suicidal attempts among sexual minorities could be attributed to the country’s cultural and political outlook. Sexual minorities in our study reported higher PHQ-9 scores. However, the odds of moderately severe to severe depression were relatively low. 

In a study by Borgogna and colleagues, sexual minorities (transgender men, women, bisexual) had mean PHQ-9 scores ranging from 10 to 13, suggesting a higher prevalence of mild to moderate depression among sexual minorities [8]. The relatively low prevalence of moderately severe to severe depression in our study can be attributed to the higher cut-off values of 15–27 that we used to define the severity level of depression in our analyses.

In a US-based study that looked at the dimensions of sexual orientation and the prevalence of mood and anxiety disorders in the US, Bostwick and colleagues found that mental health outcomes differed by sexual group, and sexual minorities were associated with higher odds of major depression [41]. The above evidence clearly indicates a strong significant association of severity of depression with sexual orientation. Future research should further explore the associations between gender, race/ethnicity, and severity of depression within the sexual minority population.

Our study had several limitations. First, we could not examine the long-term relationship between sexual orientation and depression as a result of the cross-sectional study design. Secondly, the severity of depression may have been underestimated due to misclassification issues as a result of the audio computer-assisted self-interview (ACASI) system used in the administration of the NHANES survey. Thirdly, we used self-reported PHQ-9 item scores to assess severity of depression, which does not confirm a clinical diagnosis. Fourthly, sexual minorities accounted for 5.4% of the population despite the large sample in this study. Lastly, our findings are based on the variables available in the dataset, and the observed findings could be attributed to unknown confounding.

## 5. Conclusions

Our study suggests that sexual orientation can increase the risk of depression. Moreover, lesbians (female) may have higher rates of depression compared to heterosexuals. According to the National Institute for Health and Care Excellence current clinical practice guidelines, healthcare providers are advised to pay greater attention to the symptoms of depression in people who may be at higher risk. Understanding and supporting such patient populations is important in assessing and providing effective behavioral and pharmacological treatments. Improving healthcare professionals’ ability to identify those at higher risk of having severe depression and to provide early interventions, such as referrals and resources, can reduce health disparities and improve health outcomes in this population. The association between depression and gender among sexual minorities also needs to be examined to a greater extent. Future research should explore the risk factors that contribute to health disparities among sexual minorities in detail and develop intervention opportunities to address these issues.

## Figures and Tables

**Table 1 diseases-10-00086-t001:** Comparison of demographic and socioeconomic characteristics among sexual minorities compared to heterosexuals (NHANES 2011–2016).

		Sexual Orientation		
Variables	All (N = 7826) (N, %)	Heterosexual (N = 7400) (N, %)	GLB (N = 426) (N, %)	*p*-Value
**Age in yrs. (Mean, SE)**	39.8 (0.37)	40.0 (0.28)	36.2 (0.83)	**<0.000**
**Age**				
<39 yrs.	3822 (46.0)	3545 (45.2)	277 (59.0)	**<0.001**
≥39 yrs.	4004 (54.0)	3855 (54.8)	149 (41.0)	
**Gender**				
Male	4055 (51.5)	3896 (52.1)	159 (41.9)	**0.004**
Female	3771 (48.5)	3504 (47.9)	267 (58.1)	
**Race/Ethnicity**				
Mexican American	1047 (9.1))	1009 (9.3)	38 (5.4)	**0.031**
Other Hispanic	722 (6.2)	738 (6.3)	34 (4.8)	
Non-Hispanic White	3112 (65.8)	2917 (65.6)	195 (70.5)	
Non-Hispanic Black	1773 (11.5)	1670 (11.4)	103 (11.9)	
Other Race (including Multi-racial)	1122 (7.4)	1066 (7.4)	56 (7.4)	
**Country of Birth**				
Born in the US	5857 (84.8)	5494 (84.4)	363 (91.2)	**<0.001**
Born outside the US	1965 (15.2)	1902 (15.6)	83 (8.8)	
**Citizenship Status**				
US Citizen	6781 (91.8)	6388 (91.5)	393 (95.8)	**<0.001**
Not a US Citizen	1027 (8.2)	994 (8.5)	33 (4.2)	
**Education**				
<11th Grade	1284 (12.3)	1234 (12.5)	50 (9.3)	0.141
High School Graduate/GED or Equivalent	1681 (19.9)	1579 (19.7)	102 (22.2)	
Some College or AA Degree	2648 (34.4)	2482 (34.2)	166 (38.4)	
College Graduate or above	2213 (33.4)	2105 (33.6)	108 (30.1)	
**Health Insurance**				
Yes	5783 (79.6)	5476 (79.5)	307 (80.9)	0.558
No	2038 (20.4)	1919 (20.5)	119 (19.1)	
**Poverty Income Ratio**				
<1.35	2406 (22.8)	2248 (22.2)	158 (31.6)	**0.002**
1.35–1.849	795 (9.0)	751 (9.0)	44 (9.1)	
1.85–2.99	1254 (17.4)	1180 (17.4)	74 (17.3)	
≥3	2878 (50.9)	2754 (51.4)	133 (42.0)	
**Family Income**				
<USD 25,000	2313 (23.0)	2154 (22.4)	159 (31.6)	**0.002**
USD 25,000–54,999	2153 (26.4)	2021 (26.1)	132 (31.8)	
USD 55,000–99,999	1586 (23.4)	1512 (23.7)	74 (19.0)	
≥USD 100,000	1496 (27.2)	1446 (27.8)	50 (17.6)	
**Marital Status**				
Married	3747 (52.8)	3663 (54.6)	84 (22.7)	**<0.001**
Never Married	1998 (22.5)	1790 (21.2)	208 (44.0)	
Other *	2079 (24.8)	1945 (24.2)	134 (33.3)	

GLB: gay/lesbian/bisexual; SE: standard error; bold *p*-value indicates statistical significance at <0.05; GED: General Educational Development Test; NHANES: National Health and Nutritional Examination Survey; * includes widowed, divorced, separated, and living with partner.

**Table 2 diseases-10-00086-t002:** Behavior characteristics of sexual minorities compared to heterosexuals (NHANES 2011–2016).

		Sexual Orientation		
Variables	All (N = 7826) (N, %)	Heterosexual (N = 7400) (N, %)	GLB (N = 426) (N, %)	*p*-Value
**Seen mental health professional/past year**				
Yes	742 (9.9)	657 (9.3)	85 (19.4)	**<0.001**
No	7083 (90.1)	6742 (90.7)	341 (80.6)	
**Ever told by doctor you had gonorrhea**				
Yes	29 (0.3)	27 (0.3)	2 (0.3)	0.092
No	7795 (99.7)	7371 (99.7)	424 (99.7)	
**Ever told by doctor** **you had chlamydia**				
Yes	96 (1.0)	85 (1.0)	11 (2.2)	0.054
No	7727 (99.0)	7312 (99.0)	415 (97.8)	
**Ever told by doctor** **you had genital herpes**				
Yes	333 (4.9)	300 (4.7)	33 (8.3)	**0.015**
No	7493 (95.1)	7100 (95.3)	393 (91.7)	
**Ever used cocaine/heroin/** **methamphetamine**				
Yes	1584 (22.8)	1453 (22.1)	131 (34.1)	**<0.001**
No	6230 (77.2)	5936 (77.9)	294 (65.9)	
**Ever used a needle to inject illegal drug**				
Yes	197 (2.9)	179 (2.8)	18 (3.4)	0.570
No	7625 (97.1)	7217 (97.2)	408 (96.6)	
**Do you now smoke cigarettes?**				
Every Day	1638 (41.7)	1511 (41.2)	127 (48.7)	0.171
Some Days	421(11.2)	396 (11.3)	25 (10.3)	
Not at All	1514 (47.0)	1430 (47.5)	84 (40.9)	
**Ever had 4/5 more** **alcoholic drinks?**				
Yes	1267 (16.4)	1181 (16.3)	86 (18.9)	0.332
No	6559 (83.6)	6219 (83.7)	340 (81.1)	

GLB = gay/lesbian/bisexual; bold *p*-value indicates statistical significance at <0.05; NHANES: National Health and Nutritional Examination Survey.

**Table 3 diseases-10-00086-t003:** Odds ratios of moderately severe to severe depressive symptoms on PHQ-9 items among sexual minorities compared to heterosexuals.

Item	OR (95% CI)	*p*-Value
1. Little interest in doing things	1.76 (1.22–2.54)	**0.003**
2. Feeling down, depressed, or hopeless	2.45 (1.69–3.54)	**<0.001**
3. Trouble sleeping or sleeping too much	1.30 (1.02–1.64)	**0.032**
4. Feeling tired or having little energy	1.59 (1.13–2.23)	**0.008**
5. Poor appetite or overeating	1.72 (1.20–2.47)	**0.004**
6. Feeling bad about yourself	2.95 (2.03–4.29)	**<0.001**
7. Trouble concentrating on things	2.26 (1.59–3.22)	**<0.001**
8. Moving or speaking slowly or too fast	2.41 (1.42–4.11)	**0.002**
9. Thought you would be better off dead	2.69 (1.22–5.89)	**0.015**

PHQ-9: Patient Health Questionnaire-9; OR: odds ratio; CI = confidence interval; bold *p*-value indicates statistical significance at <0.05.

**Table 4 diseases-10-00086-t004:** Regression analysis on the association of demographic characteristics with severity of depression (adjusted OR).

Variables	OR (95% CI)	*p*-Value
Sexual Minorities	1.78 (1.19–2.64)	**0.005**
Age	1.47 (1.18–1.82)	**0.001**
Gender	2.06 (1.64–2.58)	**<0.001**
Marital Status	1.23 (1.13–1.35)	**<0.001**
Education	0.76 (0.67–0.87)	**<0.001**
Family Income	0.60 (0.53–0.69)	**<0.001**
Country of Birth	0.50 (0.38–0.67)	**<0.001**

OR: odds ratio; CI = confidence interval; bold *p*-value indicates statistical significance at <0.05.

## Data Availability

The NHANES data are available in a publicly accessible repository through the Centers for Disease Control and Prevention (CDC) and the National Center for Health Statistics (NCHS). The data presented in this study are openly available on the National Health and Nutrition Examination Survey (NHANES) website at https://wwwn.cdc.gov/nchs/nhanes/ (accessed on 30 August 2021).

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
