# Peer review of "Examining Health Disparities and Severity of Depression among Sexual Minorites in a National Population Sample"

_diseases, 2022, doi:10.3390/diseases10040086_

Round 1

Reviewer 1 Report

Thank you for an interesting manuscript.  It adds to the literature about health disparities in sexual minorities by making use of a very well known representative US datasets.

Here are my specific comments.

Introduction includes a good background and relevant literature

Methods

Can the authors provide additional information about obtaining information about sensitive matters such as sexual orientation, sexual health, illicit drug use, etc.  How was this collected and then what is the risk of participants not reporting answers to these questions accurately?

It would help to know the starting sample size then how many were not included to due to a) under 18yo b) missing data (which components).  This will help the reader gauge how representative the sample remains.  Perhaps a flowchart would assist here?

The outcome of interest is self reported sexual orientation so it would be helpful to have a table that reports on the prevalence of the responses to this question.  Eg. Ln113 Proportion of Not sure Refused? 

STable 1

What does STable 1 mean?  Why is this called Table S1 at Ln177 but presented in the manuscript here.  Feels like this is supposed to be a supplementary table not in the manuscript. But it has been included in the manuscript.

The table would be better presented as N(%) as above in Table 2.  Difficult to compare N between groups.

Should the title of table 3 be "Odds ratio of moderately severe to severe..." because it is not all depression symptoms because the comparison group includes up to moderate depression symptoms?

What score from 0 (not at all) to 3 (every day) appears in this table.  Eg What score indicated that a participant had 1. Little interest in doing things?

STable 2 GLB is not in the table therefore not needed in the footnote.

Table 4 should state "sexual minority" as has previously been defined rather than sexual orientation which is not consistent with the rest of the manuscript.

Discussion

I feel like this section of the manuscript needs the most revision.

Ln214 what do you mean by poor behavioural?

Ln246 in other countries but only mentions the single country of Serbia.  Please rephrase.

Ln248 Jankovic mentioned but not included in the reference list.  My reference list only numbers to 35.

Under limitations the authors give sufficient info for the first limitation but not for the rest of the limitations.  For example at Ln261 Please describe what you mean by misclassification issues?  The reader needs some help here.

The Discussion presents lots of the data as consistent with other studies.  But what about new findings?  The information about lesbians Ln261 having more depression than heterosexuals is new info so this should be discussed and explained.

I suggest the authors revise the discussion so that each paragraph addresses a specific topic and that both consistent and new findings are discussed.

Author Response

Dear Reviewer,

Thank you for your valuable suggestions and comments. It has certainly helped us to improve our manuscript. We hope that our revision satisfies your  concerns. Appended below is the point wise response to reviewers’ comments in italics.

First Reviewer -

Thank you for an interesting manuscript.  It adds to the literature about health disparities in sexual minorities by making use of a very well known representative US datasets.

We thank reviewer for his/her appreciation of our research 

Here are my specific comments.

Introduction includes a good background and relevant literature

Thank you so much for the compliment.

Methods

Can the authors provide additional information about obtaining information about sensitive matters such as sexual orientation, sexual health, illicit drug use, etc.  How was this collected and then what is the risk of participants not reporting answers to these questions accurately?

Certainly. The sexual behavior questions were self-administered in a private room at the Mobile Examination Centers (MEC) using the Audio Computer-Assisted Self Interview (ACASI) system. The interview was conducted in one of the following languages: English, Spanish, Korean, Vietnamese, or Chinese (traditional/Mandarin, simplified/Mandarin, and traditional/Cantonese). The ACASI enables respondents to hear questions through earphones as well as read questions on the computer screen. Respondents move at their own speed and touch the screen to indicate their response. No proxy respondents or translators were used in situations when the respondents could not self-report. Respondents had option to “refuse” or “would rather not answer” option if they felt uncomfortable answering or choose not to answer at all. There was minimal risk (emotional distress) to the participants in answering or not answering such sensitive questions. This information is now added to the manuscript to provide a greater clarity.      

It would help to know the starting sample size then how many were not included to due to a) under 18yo b) missing data (which components).  This will help the reader gauge how representative the sample remains.  Perhaps a flowchart would assist here?

We agree with the reviewer’s comment. We have added this information to the manuscript to provide greater clarity.

The outcome of interest is self reported sexual orientation so it would be helpful to have a table that reports on the prevalence of the responses to this question.  Eg. Ln113 Proportion of Not sure Refused? 

We agree with the reviewer’s comment. We have added this information to the manuscript to provide greater clarity.

STable 1

What does STable 1 mean?  Why is this called Table S1 at Ln177 but presented in the manuscript here.  Feels like this is supposed to be a supplementary table not in the manuscript. But it has been included in the manuscript.

We sincerely apologize for this confusion. However, this was done in accordance with the journal’s formatting requirement that tables need to be placed where they are referred in the text of the manuscript. However, we are assuming that at the time of publication, this information will be made available separately as a supplementary material and will not be included as a part of the manuscript.  

The table would be better presented as N(%) as above in Table 2.  Difficult to compare N between groups.

We agree with the reviewer. The table is now updated with n, % for each group.

Should the title of table 3 be "Odds ratio of moderately severe to severe..." because it is not all depression symptoms because the comparison group includes up to moderate depression symptoms?

We agree with the reviewer and have modified the title of Table 3 as suggested.  

What score from 0 (not at all) to 3 (every day) appears in this table.  Eg What score indicated that a participant had 1. Little interest in doing things?

We apologize for this confusion and lack of clarity. As stated in our methods section. The PHQ-9 questionnaire includes 9 items including an item little interest in doing things (actual wording of the item: Over the last 2 weeks, how often have you been bothered by the following problems: little interest or pleasure in doing things?). All of the 9 items of PHQ-9  were scored on a scale from 0 (not at all) to 3 (nearly every day). In this study, we defined the presence of none to moderate depression (a score of 0-14) and moderately severe to severe depression (a score of 15-27).

STable 2 GLB is not in the table therefore not needed in the footnote.

We agree with the reviewer. It is now removed from the footnote

Table 4 should state "sexual minority" as has previously been defined rather than sexual orientation which is not consistent with the rest of the manuscript.

We agree with the reviewer. It is now replaced with word “sexual minorities”  for consistency.  

Discussion

I feel like this section of the manuscript needs the most revision.

Ln214 what do you mean by poor behavioural?

We apologize for a missing word between poor and behavior. It should have been poor health behavior – as smoking, heavy drinking, illicit drug use, substance use etc. We have replaced this word with “poor health behavior”

Ln246 in other countries but only mentions the single country of Serbia.  Please rephrase.

We thank reviewer for pointing this out. We have rephrased this sentence accordingly

Ln248 Jankovic mentioned but not included in the reference list.  My reference list only numbers to 35.

We thank reviewer for pointing out this glaring error. We have now added this and other missing citations to our references   

Under limitations the authors give sufficient info for the first limitation but not for the rest of the limitations.  For example at Ln261 Please describe what you mean by misclassification issues?  The reader needs some help here.

We apologize for the lack of clarity. We have rephrased this section to provide greater clarity. Misclassification (differential and non-differential) is an error that can occur at any stage of the research process when an individual inadvertently included in a different group than the one to which they should have been included because of some kind of observational or measurement error.

Although the Audio Computer-Assisted Self Interview (ACASI) system used in NHANES has many advantages over interviewer-administered questionnaires for sensitive information, inconsistent responses can occur. Since the questions are self-administered, unlikely answers or confusing questions generally cannot be clarified by the interviewer. Although, the questionnaire was programmed to alert respondents of potential data entry errors or inconsistencies, not every possible consistency check was added. To minimize potential respondent frustration and confusion, the interview was not interrupted each time an inconsistent answer occurred in terms of the number of sexual partners for different periods of recall for different types of sexual behavior (Source: NHANES data documentation and codebook).

The Discussion presents lots of the data as consistent with other studies.  But what about new findings?  The information about lesbians Ln261 having more depression than heterosexuals is new info so this should be discussed and explained.

We agree with the reviewer. We have modified and rephrased our discussion accordingly to discuss new information. 

I suggest the authors revise the discussion so that each paragraph addresses a specific topic and that both consistent and new findings are discussed.

We thank reviewer for pointing this out. We have modified the discussion accordingly to address specific topic, compare and contrast our findings with the published literature and discuss new findings. In addition, we have included few more new references.    

Reviewer 2 Report

In this manuscript, the authors examined health disparities and severity of depression among sexual minorities in national population samples. This is a well-written manuscript. I have the following minor comments. 

1. Please use the Chi-squared test instead of the Chi-square test. 

2. The authors should provide more details for data analysis. How do you use SPSS and STATA to account for sampling weights and the complex nature of sampling design? 

3. The data are clearly imbalanced for your multivariate logistic regression model. Did you try weighted-logistic regression to handle the imbalance issues in your data? 

Author Response

Dear Reviewer,

Thank you for your valuable suggestions and comments. It has certainly helped us to improve our manuscript. We hope that our revision satisfies your concerns. Please refer to the attached file for our response. 

Second Reviewer-

In this manuscript, the authors examined health disparities and severity of depression among sexual minorities in national population samples. This is a well-written manuscript. I have the following minor comments. 

We sincerely thank reviewer for his/her compliment on our manuscript. 

  1. Please use the Chi-squared test instead of the Chi-square test. 

We thank reviewer for pointing out this error. This error has been corrected now.

  1. The authors should provide more details for data analysis. How do you use SPSS and STATA to account for sampling weights and the complex nature of sampling design? 

We apologize for the lack of clarity. We have used the SPSS for data cleaning, merging, management of the data and analyses of unweighted sample. Whereas STATA was used to perform all statistical analyses with weighted sample. STATA provides the functionality to incorporate primary sampling units, strata and sampling weights (sampling design) to estimate the standard errors and point estimates accurately. We have included this information in the methods section to provide more clarity.  

Weighting of the NHANES data produces estimates representative of the civilian resident noninstitutionalized U.S. population. The weighting of sample data permits analysts to produce estimates of the statistics that would have been obtained if the entire eligible population had been surveyed. Sample weights are considered measures of the number of persons in the target population represented by the particular participant. Weighting considers several features of the survey design: the differential probabilities of selection for the sampling domains, survey nonresponse, and differences between the final sample distribution and the target population distribution (Source: NHANES analytical guidelines).

  1. The data are clearly imbalanced for your multivariate logistic regression model. Did you try weighted-logistic regression to handle the imbalance issues in your data? 

We thank reviewer for pointing this out and for his/her valuable suggestion. Such imbalances in data are usually handled with methods like SMOTE, class weights or weighted logistics regression as suggested most commonly used in machine learning. We will keep this in mind in our future research. As stated in our manuscript, we have used logistic regression model to quantify the magnitude of association between severity of depression and sexual orientation. It is not used as a prediction model  or to build a prediction model. All of our analyses were performed using the sampling weights as suggested by the NHANES analytical guidelines. In response to your suggestion, we re-ran the model and plotted area under the curve (AUC) for our logistic regression. The AUC was 75.5% suggesting that the model had few wrong predictions in majority (heterosexuals) class and fair predictability for minority class (sexual minorities).